# Pleckstrin homology domain-containing protein PHLDB3 supports cancer growth via a negative feedback loop involving p53

Tengfei Chao[1,2,3], Xiang Zhou[1,2,†], Bo Cao[1,2], Peng Liao[1,2], Hongbing Liu[1,†], Yun Chen[1], Hee-Won Park[1], Shelya X. Zeng[1,2] & Hua Lu[1,2]

The tumour suppressor p53 transactivates the expression of its target genes to exert its functions. Here, we identify a pleckstrin homology domain-containing protein (PHLDB3)-encoding gene as a p53 target. PHLDB3 overexpression increases proliferation and restrains apoptosis of wild-type p53-harboring cancer cells by reducing p53 protein levels. PHLDB3 binds to MDM2 (mouse double minute 2 homolog) and facilitates MDM2-mediated ubiquitination and degradation of p53. Knockdown of PHLDB3 more efficiently inhibits the growth of mouse xenograft tumours derived from human colon cancer HCT116 cells that contain wild type p53 compared with p53-deficient HCT116 cells, and also sensitizes tumour cells to doxorubicin and 5-Fluorouracil. Analysis of cancer genomic databases reveals that PHLDB3 is amplified and/or highly expressed in numerous human cancers. Altogether, these results demonstrate that PHLDB3 promotes tumour growth by inactivating p53 in a negative feedback fashion and suggest PHLDB3 as a potential therapeutic target in various human cancers.

[1] Department of Biochemistry & Molecular Biology, Tulane University School of Medicine, New Orleans, Louisiana 70112, USA. [2] Tulane Cancer Center, Tulane University School of Medicine, New Orleans, Louisiana 70112, USA. [3] Department of Oncology, Tongji Hospital, Tongji Medical College, Huazhong University of Science and Technology, Wuhan 430030, China. † Present addresses: Shanghai Cancer Center and Institutes of Biomedical Sciences, Fudan University, Shanghai 200032, China (X.Z.); Department of Pediatrics, Tulane University School of Medicine; New Orleans, Louisiana 70112, USA (H.L.). Correspondence and requests for materials should be addressed to H.L. (email: hlu2@tulane.edu).

The p53 tumour suppressor plays a crucially important role in preventing cancer development[1] as strongly supported by the facts that Trp53-null mice develop cancer in 100% penetrance, and its gene is mutated in over half of all human cancers[1,2]. In response to a variety of stresses, the p53 transcriptional regulator can induce or repress transcription of numerous target genes, which are involved in multiple biological functions, including cell cycle, apoptosis, cell senescence, differentiation, angiogenesis, cell migration, metabolism and DNA repair[3]. For example, the p53 target genes CDKN1A (p21) and GADD45 are involved in p53-dependent cell cycle arrest, while the BH3-only-encoding target genes BBC3 (Puma) and PMAIP1 (Noxa) play key roles in p53-mediated apoptosis[4]. Although a number of target genes involved in p53-dependent cell cycle arrest and apoptosis have been well characterized, the p53-regulated network in these processes is not completely understood[5]. Therefore, identification of additional p53 target genes will further update our knowledge about how p53 acts as a vital tumour suppressor under different cellular stresses.

Due to its detrimental effect on cells, p53 is often inactivated in human cancers that harbor wild type TP53 (refs 1,2,6). Under physiological conditions, p53 is maintained at an extremely low level due to its rapid ubiquitination-dependent proteasomal degradation mediated by MDM2 (mouse double minute 2 homolog), which possesses an E3 ubiquitin ligase activity. MDM2 often works together with MDMX (also known as MDM4) to negatively regulate the stability and activity of p53 protein in a feedback fashion[7-9]. Interestingly, besides MDMX, other proteins have been suggested to modulate Mdm2-mediated p53 ubiquitination and degradation, including Yin-Yang1 (ref. 10), gankyrin[11], Daxx[12] and our recently identified NGFR[13]. However, it remains to be found out if there are still more yet unidentified regulators of this feedback loop.

Our latest study as presented here surprisingly unraveled PHLDB3 (pleckstrin homology-like domain, family B, member 3; also known as LL5γ) as another feedback regulator of p53. PHLDB3 is a member of the LL5 family, which consists of PHLDB1, -2, and -3 (or LL5α, -β, and -γ). It is also an understudied protein containing two predicted coiled-coil domains, and a phosphositide-binding module called PH domain, as little has been known about this protein. As the shortest protein in the family, PHLDB3 shares 30–48% similarity in coiled-coil domains and 76–78% in PH domain with PHLDB1 and PHLDB2 (ref. 14). Though the function of LL5 protein family is underappreciated and barely studied, evidence has been gradually accumulated to unveil the cellular functions of PHLDB1 and PHLDB2. For example, PHLDB2 has been shown as a microtubule-anchoring factor that binds with CLASP involved in the interaction between distal microtubule ends and the cell cortex[14]. PHLDB1 and PHLDB2 work together to play a role in laminin-dependent microtubule anchoring at the epithelial cell basal cortex[15]. In addition, PHLDB1 can bind PI (3, 4, 5) P3 through its PH domain in adipocytes and function as a positive regulator of Akt activation by insulin[16]. More recent studies[17-19] have shown the correlation between PHLDB1 SNPs and glioma risk, implicating that PHLDB1 may play a potential role in the development of glioma. Noteworthily, PHLDA3, one of the pleckstrin homology-like domain family A proteins, was previously reported as a direct target gene of p53 (ref. 20). As a PH domain-only protein, PHLDA3 suppresses Akt activity by competing with Akt for binding to membrane lipids and functions as a tumour suppressor in pancreatic neuroendocrine tumours[21]. However, to date, little attention has been paid to the physiological or pathological functions of PHLDB3.

Our study as detailed below also reveals PHLDB3 as another direct target of p53. Different from PHLDA3 (ref. 21), PHLDB3 interacts with MDM2 and promotes MDM2-dependent ubiquitination and proteasomal degradation of p53. Moreover, ectopic PHLDB3 promotes cell growth and inhibits apoptosis in both cell-based and xenograft tumour models. Inversely, knockdown of this protein induces apoptosis and inhibits cell growth, which is more significant in p53 wild-type-containing cancer cells. Interestingly, PHLDB3 is amplified and/or upregulated in a number of human cancers, and the high expression of PHLDB3 is significantly correlated with rare p53 mutations in some breast or esophageal cancers. Therefore, our findings demonstrate that PHLDB3 can play an oncogenic role by at least in part inhibiting p53 activity in a feedback fashion.

## Results

**p53 is required for PHLDB3 induction by cytotoxic drugs.** In our previous studies to investigate the mechanisms of Inauhzin (INZ) on the p53 pathway in cancer cells[22-24], we identified PHLDB3 as a potential p53-regulated gene. To confirm this result, we treated HCT116, H1299 and H460 cells with Doxorubicin (Dox) or 5-Fluorouracil (5-Fu) and found that these chemotherapeutic drugs increase the mRNA level of PHLDB3 dramatically in wild-type p53-containing cells, including HCT116$^{p53+/+}$ and H460 (Fig. 1a), but not in p53-null cells (HCT116$^{p53-/-}$ or H1299). Consistently, the protein level of PHLDB3 was also induced in wild-type p53-containing cells, but not p53-null cells, after the drug treatments (Fig. 1b–d). Inversely, knockdown of p53 using siRNA in HCT116$^{p53+/+}$ and H460 cells resulted in marked decrease of Dox- or 5-Fu-induced mRNA and protein levels of PHLDB3 (Fig. 1e–h). These results are statistically significant and indicate that PHLDB3 induction by chemotherapeutic drugs is p53 dependent, and therefore this gene might be another p53 target.

**p53 directly regulates PHLDB3 expression.** To validate if PHLDB3 indeed is a p53 target gene, we overexpressed p53 in p53-null cells, H1299. As shown in Fig. 2a,b, ectopic p53 significantly induced both of the PHLDB3 mRNA and protein levels. Also, using the p53MH algorithm program, we identified two potential p53 responsive elements (RE1 and RE2) upstream from the transcriptional initiation site of the human PHLDB3 gene on chromosome 19 (Fig. 2c). Ectopic p53 induced the activity of a luciferase reporter, whose expression was driven by the PHLDB3 promoter that contains RE1, but not RE2 or mutated RE1 (Fig. 2d), suggesting that RE1 must be the p53-binding DNA element in this promoter. This result was further verified by chromatin-associated immunoprecipitation (ChIP) assays, which showed that endogenous p53 specifically binds to the RE1-containing PHLDB3 promoter, but not the RE2-containing promoter, in response to Dox treatment in HCT116$^{p53+/+}$ cells (Fig. 2e). Taken together, these results demonstrate that PHLDB3 is an authentic p53 transcriptional target gene.

**PHLDB3 promotes cancer cell growth and inhibits apoptosis.** Because little is known about the biological and biochemical functions of PHLDB3, we first searched several available genomic and gene expression databases for PHLDB3 status. Interestingly, our analysis of TCGA genome database[25,26] revealed that the PHLDB3 gene is extensively amplified in a variety of cancers, including pancreas, uterine, bladder, breast, lung, colorectal, sarcoma and liver cancers (Fig. 3a). In line with this result, our analysis of Oncomine database[27] also showed that the PHLDB3 mRNA expression is significantly higher in breast and

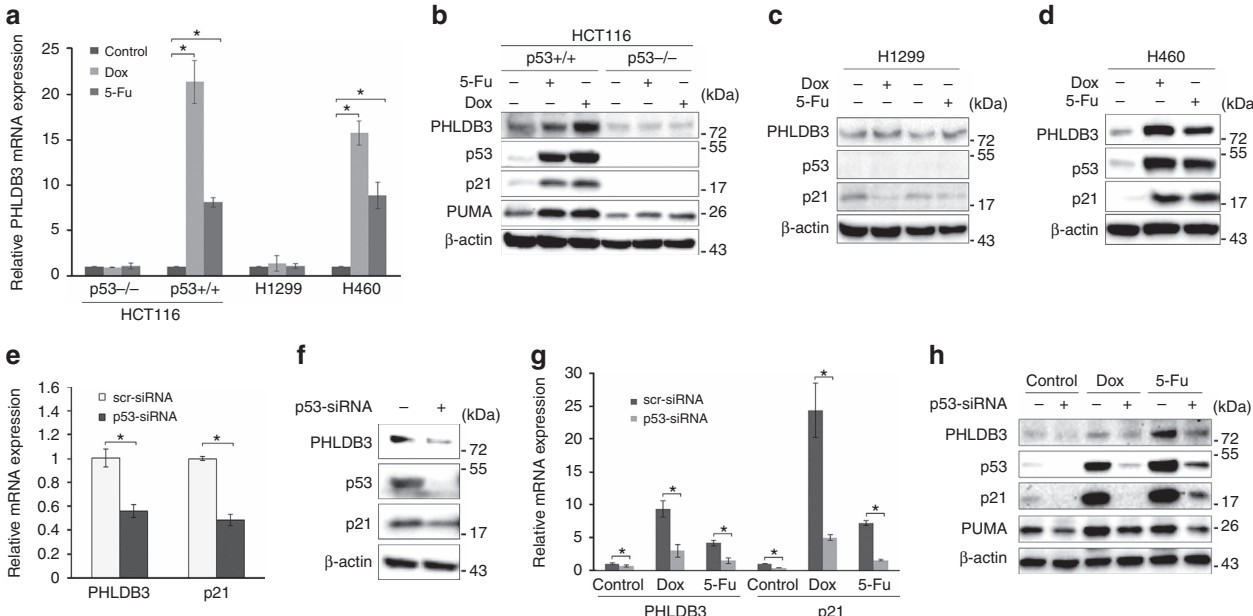

**Figure 1 | PHLDB3 is induced by chemotherapy drugs in a p53-dependent manner.** (a–d) Analysis of PHLDB3 mRNA and protein levels following chemotherapy drugs treatment. HCT116, H1299 and H460 cells were treated with Doxorubicin (Dox) or 5-Fluorouracil (5-Fu) for 16 h for analyses of RNA and protein levels. The mRNA levels of PHLDB3 were measured using RT-qPCR (a). The protein levels of PHLDB3, p53 and p53 targets were detected using immunoblotting analysis with indicated antibodies (b–d). (e,f) Analysis of PHLDB3 mRNA and protein levels after p53 knockdown. H460 cells were transfected with p53 or scramble siRNA, and cells were harvested 72 h post-transfection for RT-qPCR (e) or immunoblotting with indicated antibodies (f). (g,h) The effect of p53 knockdown on the PHLDB3 mRNA (g) and protein levels (h) after treatment of HCT116$^{p53+/+}$ cells with chemotherapy drugs. HCT116$^{p53+/+}$ cells were transfected with p53 or scramble siRNA for 72 h, and treated with Dox or 5-Fu for 16 h before the cells were harvested for RT-qPCR (g) or immunoblotting with indicated antibodies (h). Where applicable, data were corrected for GAPDH and represent mean ± s.e.m. of triplicate experiments. *$P < 0.05$ by two-tailed $t$-test (a,e,g).

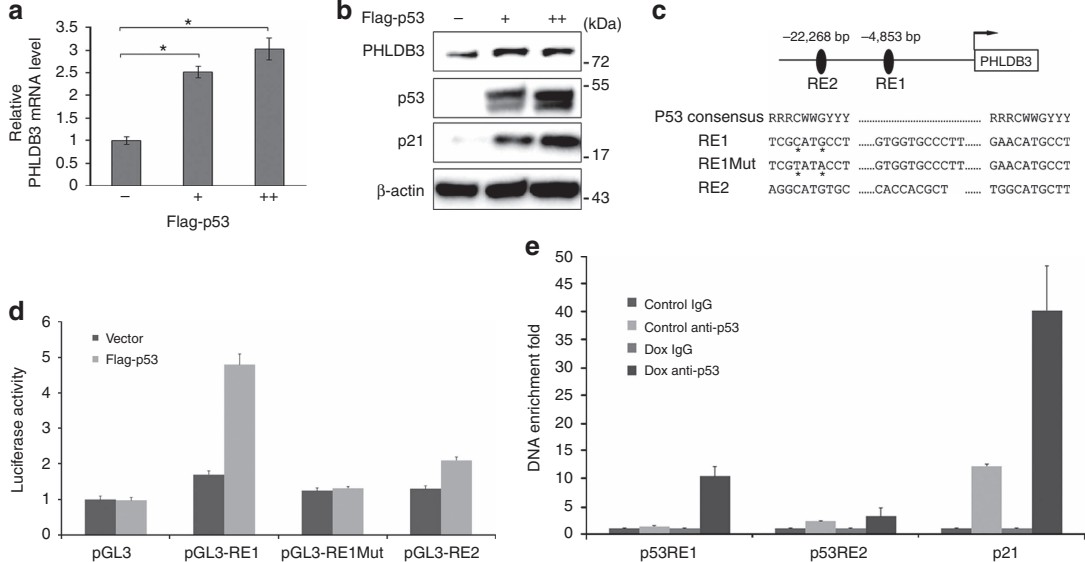

**Figure 2 | PHLDB3 is a direct transcriptional target gene of p53.** (a,b) The effect of p53 overexpression on PHLDB3 mRNA (a) and protein levels (b). H1299 cells were transfected with p53 and harvested 36 h post-transfection. The relative PHLDB3 mRNA level was quantified by RT-qPCR. Data was corrected for GAPDH and represent mean ± s.e.m. of triplicate experiments. *$P < 0.01$ by two-tailed $t$-test (a). The PHLDB3, p53 and p21 levels were determined by immunoblotting with corresponding antibodies (b). (c) Two potential p53 responsive elements (RE1 and RE2) were identified in the human PHLDB3 promoter region using computer software (p53MH algorithm). The CATG to TATA mutation of RE1 was generated by site-directed mutagenesis. (d) p53 induces the activity of luciferase, whose expression was driven by the RE1, but not RE2 or mutated RE1, in the PHLDB3 promoter. Data represent mean ± s.e.m. of triplicate experiments.*$P < 0.01$ by two-tailed $t$-test. (e) p53 binds to the PHLDB3 promoter in cells. HCT116$^{p53+/+}$ cells were treated with 1 μM DOX for 16 h, and ChIP assays were performed with the p53 antibody or control IgG. The promoter regions of indicated genes were analysed by RT-qPCR. *$P < 0.01$ by $t$-test.

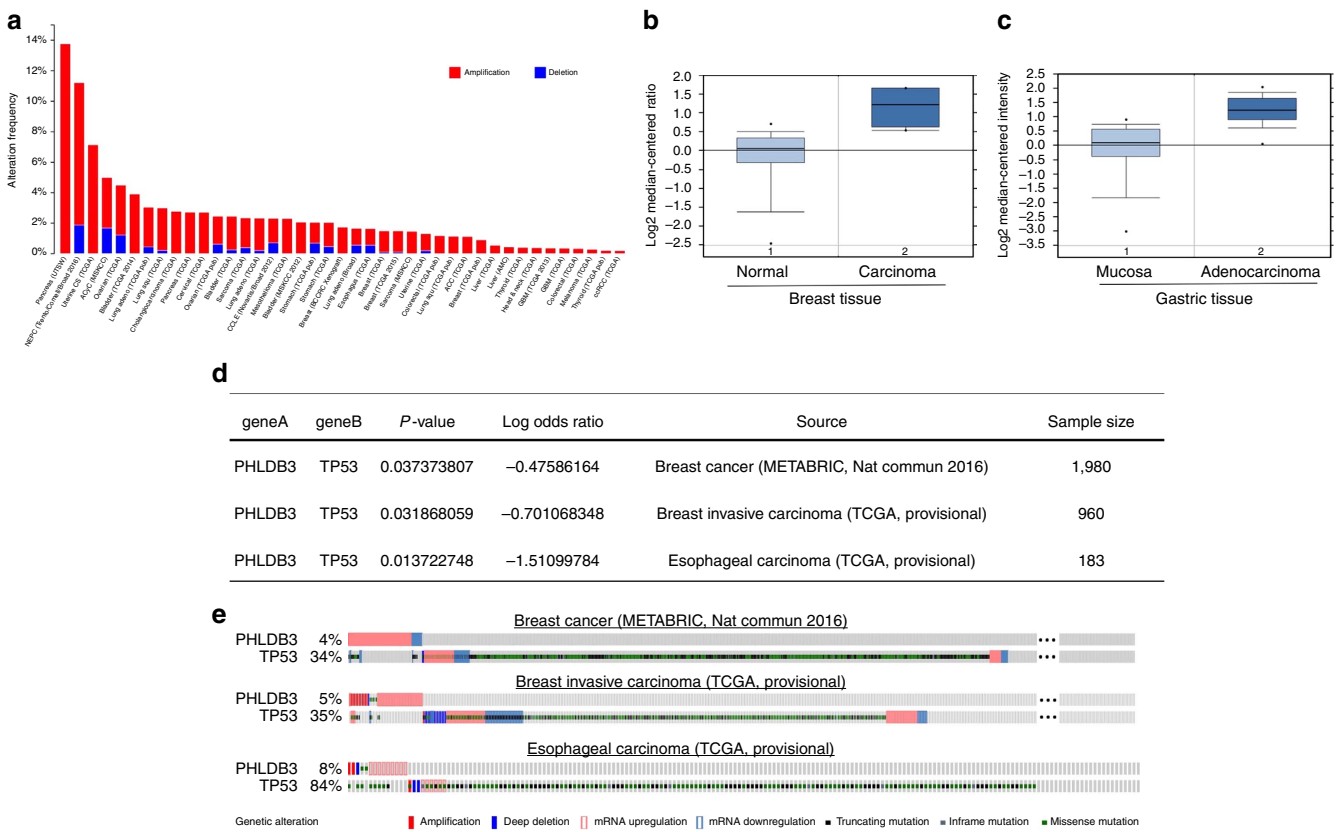

**Figure 3 | High expression of PHLDB3 is found in multiple cancers and mutually exclusive to TP53 mutations in some cancers.** (**a**) TCGA database was searched, and the data were modified from the cBioPortal for Cancer Genomics (http://www.cbioportal.org/). (**b,c**) The expression profile of PHLDB3 in cancers and normal tissues was searched in Oncomine Gene Browser (http://www.oncomine.org/). The results in TCGA Breast Dataset (**b**) and DErrico Gastric Dataset (**c**) are shown. (**d**) Analysis of mutual exclusivity was retrieved from cBioPortal and gene alteration status of PHLDB3 and TP53 of individual sample is depicted in **e**.

gastric cancers[28] than in their normal tissues (2.497 and 2.616 folds, respectively) (Fig. 3b,c). Remarkably, our further relevance analysis of gastric cancer database showed that patients with higher PHLDB3 expression display a much shorter disease-free survival rate than patients with lower PHLDB3 expression (Supplementary Fig. 1). Intriguingly, upregulation of PHLDB3 was significantly correlated with rare TP53 mutations in breast cancer, breast invasive carcinoma and esophageal carcinoma as revealed by the analysis of cBioPortal databases[25,26] (Fig. 3d,e). These results suggest that PHLDB3 might play an oncogenic role in human cancer development, potentially through inactivation of p53.

To test this possibility, we first investigated whether down-regulation of PHLDB3 expression by siRNA would affect cancer cell apoptosis. Interestingly, knockdown of PHLDB3 led to more marked apoptosis of HCT116$^{p53+/+}$ cells than that of HCT116$^{p53-/-}$ cells, as measured by sub-G1 population and cleaved PARP (Fig. 4a,b). Since the effect of PHLDB3 knockdown on apoptosis appeared to be more significant in p53-containing cancer cells, we then analysed the expression of p53 and its known target genes. Surprisingly, knockdown of PHLDB3 led to marked induction of p53 protein level, but not mRNA level (Fig. 4c,d), and consequent induction of the protein and mRNA levels of several of its target genes, such as Puma, p21 and MDM2 (Fig. 4c,d), suggesting that PHLDB3 might be involved in regulation of p53 pathway. Consistent with this speculation, overexpression of this gene in HCT116$^{p53+/+}$ cells markedly reduced the protein levels (Fig. 4f) of p53 and its targets, such as Puma and p21, and the mRNA levels (Fig. 4e) of these target

genes, but had no effect of p53 mRNA level (Fig. 4e). To further confirm the effect of PHLDB3 on cancer cell survival, we generated HCT116 cell lines that stably express PHLDB3 shRNA. As shown in Fig. 4g, the expression of p53 and some of its target genes was markedly induced by PHLDB3 shRNA in HCT116$^{p53+/+}$ cells, but not in HCT116$^{p53-/-}$ cells. As anticipated above (Fig. 3), PHLDB3 silencing led to marked reduction of colon cancer cell viability, which was more significant in HCT116$^{p53+/+}$ cells (Fig. 4i) than in HCT116$^{p53-/-}$ cells (Fig. 4h). Similar observation of apoptosis induction and cell viability suppression by PHLDB3 knockdown were also found in lung cancer H460 (wild-type p53) and H1299 (p53 null) cells (Supplementary Fig. 2), indicating that the oncogenic properties of PHLDB3 are not limited to colon cancer cells. Using colon cancer cell lines for colongenic assays, we found that knockdown of PHLDB3 lead to statistically significant reduction of colony formation in both p53-containing and p53-deficient HCT116 cells, but clearly more dramatically in p53-positive colon cancer cells (Fig. 4j,k). Collectively, these results demonstrate that PHLDB3 plays a crucial role in cancer cell survival by predominantly suppressing the p53 pathway, though this oncoprotein might also possess p53-independent functions in regulation of cell growth and survival.

**PHLDB3 promotes p53 degradation by enhancing ubiquitination.** Because either knockdown or overexpression of PHLDB3 only affected the protein level of p53 (Fig. 4e–h), we next sought to determine if PHLDB3 could affect MDM2-mediated p53

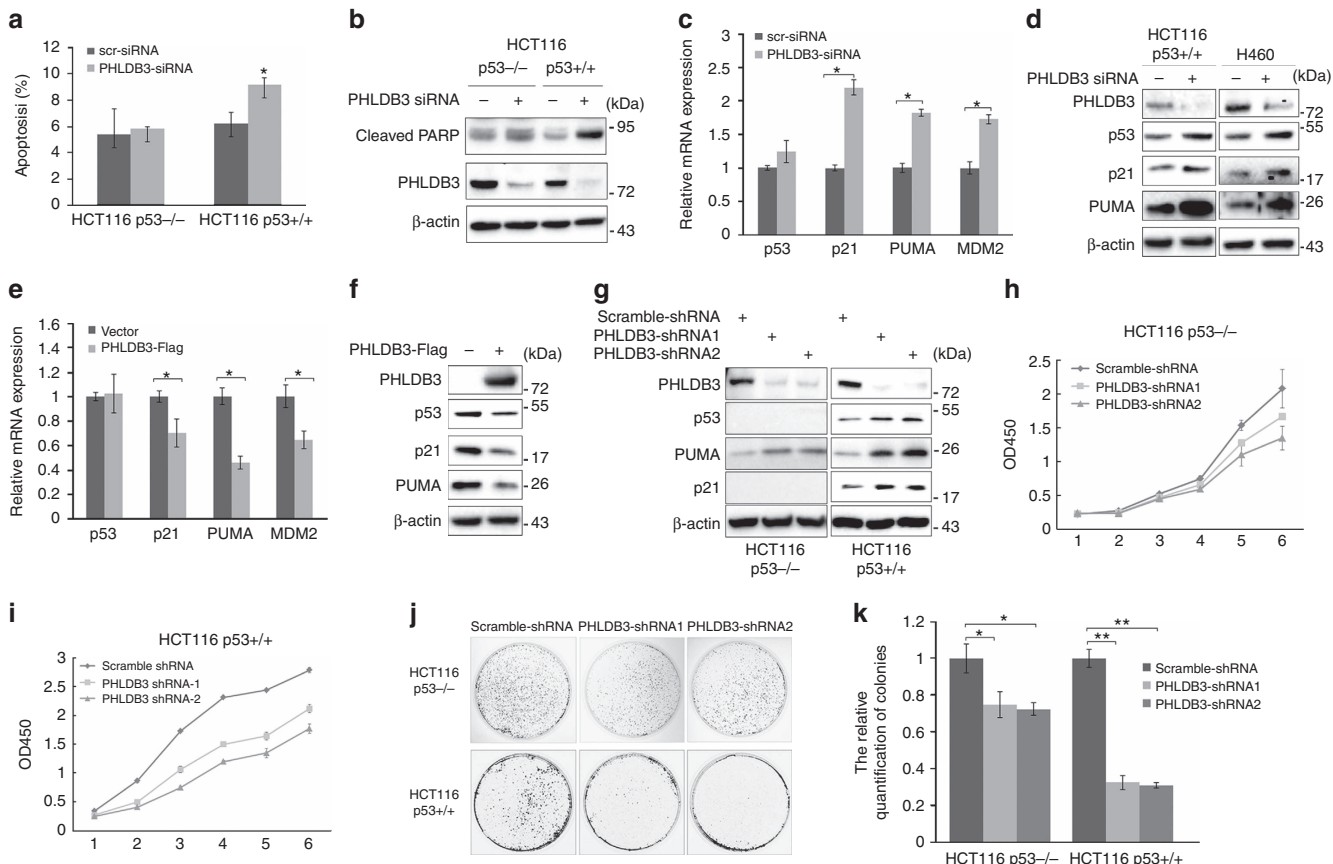

**Figure 4 | PHLDB3 knockdown inhibits cell proliferation and promotes apoptosis of colon cancer cells by inducing p53 protein level.** (**a,b**) The effect of PHLDB3 knockdown on apoptosis of HCT116$^{p53-/-}$ and HCT116$^{p53+/+}$ cells. HCT116$^{p53-/-}$ and HCT116$^{p53+/+}$ cells were transfected with PHLDB3 or scramble siRNA and harvested 72 h post-transfection for flow cytometry analysis (**a**) or immunoblotting with indicated antibodies (**b**). Quantification of Sub-G1 population is shown in **a**. (**c–f**) Knockdown of PHLDB3 induces, but overexpression of PHLDB3 reduces, p53 and its target genes. HCT116$^{p53+/+}$ (**c,d**) and H460 (**d**) cells were transfected with PHLDB3 or scramble siRNA and harvested 72 h post transfection for RT-qPCR (**c**) or immunoblotting with indicated antibodies (**d**). HCT116$^{p53+/+}$ cells were transfected with PHLDB3-Flag or vector plasmid and harvested 48 h post transfection for RT-qPCR (**e**) or immunoblotting with indicated antibodies (**f**). (**g**) Knockdown of PHLDB3 causes p53-dependent induction of p21 and Puma. The protein levels of p53 and its targets, p21 and Puma, in HCT116$^{p53-/-}$ and HCT116$^{p53+/+}$ cells that stably expressed PHLDB3 or scramble shRNA were detected by immunoblotting using antibodies as indicated. (**h,i**) The effect of PHLDB3 knockdown on colon cancer cell growth. HCT116$^{p53-/-}$ (**h**) and HCT116$^{p53+/+}$ (**i**) cells that stably expressed PHLDB3 or scramble shRNA were seeded in 96-well plate and cell viability was evaluated every 24 h by CCK8. (**j,k**) Knockdown of PHLDB3 leads to inhibition of clonogenic capability of colorectal cancer cells, more significantly when the cells harbor wild type p53. HCT116$^{p53-/-}$ and HCT116$^{p53+/+}$ cells that stably expressed PHLDB3 shRNA were seeded on 10-cm plates for 10–14 days, and colonies were fixed by methanol and stained with crystal violet solution (**j**). The relative quantification of colonies is shown in **k**. Where applicable, data represent mean ± s.e.m. of triplicate experiments.*$P<0.05$, **$P<0.01$ by two-tailed $t$-test (**a,c,e,k**).

ubiquitination as this is the key mechanism responsible for p53 turnover[7]. Indeed, this was the case, as ectopic PHLDB3 enhanced MDM2-mediated p53 ubiquitination (Fig. 5a). Consistently, co-expression of PHLDB3 with MDM2 led to more reduction of p53 protein levels, which was abolished by proteasome inhibitor MG132 (Fig. 5b). In contrast, knockdown of PHLDB3 resulted in marked increase of p53's half-life, as detected by immunoblotting assays following the treatment of HCT116 cells with cyclohexamide, a translation inhibitor (Fig. 5c,d). The induction of p53 degradation by PHLDB3 was MDM2 dependent, as overexpression of this protein alone in p53 and MDM2 double knockout MEF cells had no effect on the protein level of ectopic p53 (Fig. 5e). Taken together, these results demonstrate that PHLDB3 can reduce p53 stability by enhancing MDM2-mediated p53 ubiquitination and degradation.

**PHLDB3 interacts with MDM2.** Additional evidence supporting the functional cooperation between PHLDB3 and MDM2 was

obtained by co-immunoprecipitation (co-IP) assays. First, we carried out co-IP assays by overexpressing both PHLDB3-FLAG and HA-MDM2 in HEK293 cells and found that these two proteins are co-immunoprecipitated with each other in reverse co-IP assays (Fig. 6a,b). Their interaction was further verified by analyzing their protein complex with endogenous PHLDB3 and MDM2 in HCT116$^{p53+/+}$ cells after Dox treatment (Fig. 6c). As shown in Fig. 6d, recombinant His-PHLDB3 and GST-MDM2 purified from bacteria could also form a complex in a GST-fusion protein–protein interaction assay, suggesting direct interaction of these two proteins. Interestingly, different from other MDM2-binding proteins, such as ribosomal proteins[29,30], PHLDB3 preferred to binding to the C-terminal domain of MDM2, as PHLDB3 was specifically pulled down with the C-terminal (aa 284–491) fragment, but not N-terminal fragments, of GST-tagged MDM2, though to a lesser degree (Fig. 6d). Since MDMX also binds to MDM2 through their C-termini, we then tested whether PHLDB3 could affect the interaction between MDM2 and MDMX. The result surprisingly showed that ectopic

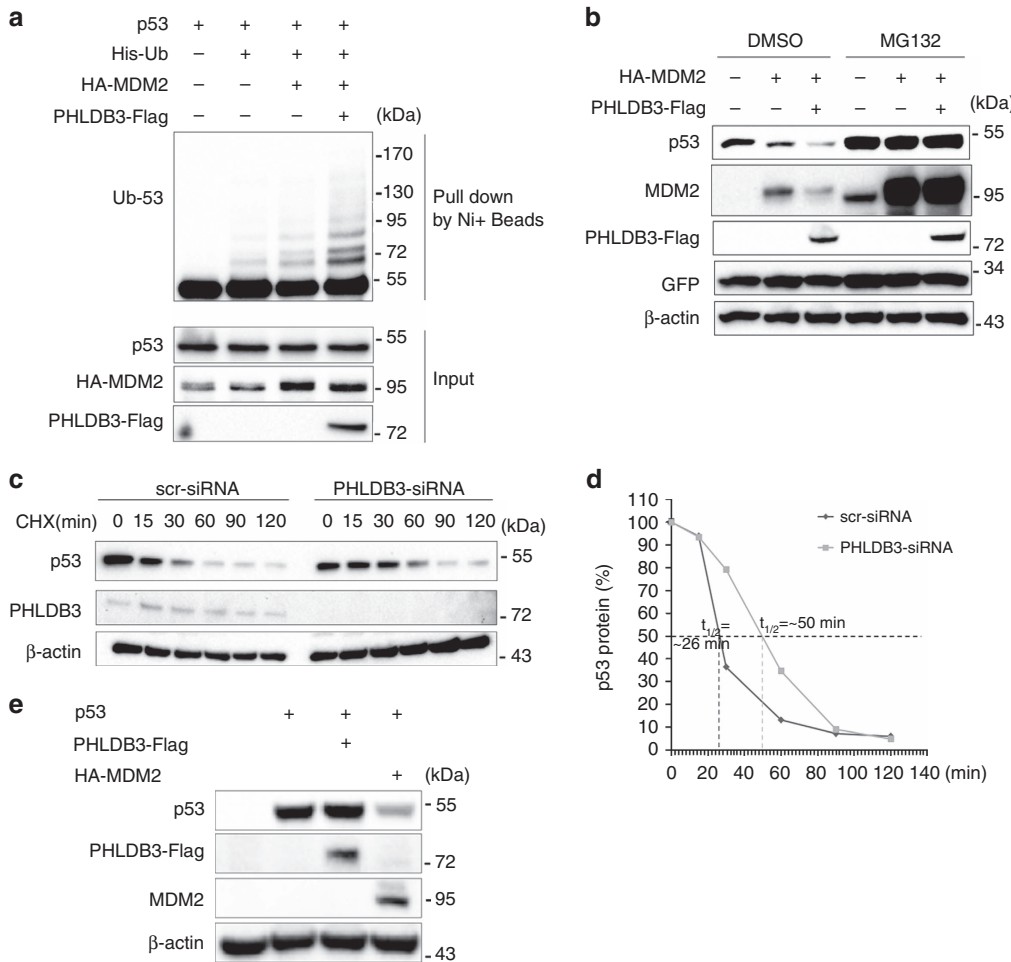

**Figure 5 | PHLDB3 promotes p53 degradation by enhancing MDM2-mediated ubiquitination. (a)** PHLDB3 promotes MDM2-induced p53 ubiquitination. HCT116$^{p53-/-}$ cells were transfected with combinations of plasmids encoding p53, HA-MDM2, PHLDB3-Flag or His-Ub, and treated with MG132 6 h before harvested for *in vivo* ubiquitination assay. Bound and input proteins were detected by immunoblotting using antibodies as indicated. **(b)** PHLDB3 enhances MDM2-mediated p53 proteasomal degradation. HCT116$^{p53+/+}$ cells were transfected with combinations of plasmids encoding EGFP, HA-MDM2 or PHLDB3-Flag followed by immunoblotting using antibodies as indicated. MG132 was supplemented to the medium for 6 h. **(c,d)** p53-half-life is increased upon PHLDB3 knockdown. HCT116$^{p53+/+}$ cells transfected with PHLDB3 or scramble siRNA for 72 h were treated with 100 µg ml$^{-1}$ of CHX and harvested at different time points as indicated. The p53 protein level was detected by immunoblotting **(c)**, quantified by densitometry and plotted against time to determine p53-half-lives **(d)**. **(e)** Ectopically expressed PHLDB3 does not alter p53 protein level in the absence of MDM2. MEF$^{p53-/-; Mdm2-/-}$ cells were transfected with combinations of plasmids encoding PHLDB3-Flag, HA-MDM2 or p53 followed by immunoblotting using antibodies as indicated.

PHLDB3 reduces the amount of MDMX–MDM2 complexes in a dose-dependent manner as measured by a co-IP assay (Supplementary Fig. 3a). Interestingly, PHLDB3 still interacted with C464A MDM2 mutant (Supplementary Fig. 3b), which failed to bind to MDMX (Supplementary Fig. 3c) as expected. These results suggest that PHLDB3 could compete with MDMX for binding to the RING domain of MDM2, although they appeared to bind to different residues within the Ring domain of MDM2. Moreover, ectopic PHLDB3 did not apparently affect the auto-ubiquitination and half-life of MDM2 (Supplementary Fig. 4). Using a set of co-IP-IB assays with ectopically expressed HA-MDM2 and PHLDB3 fragments tagged with Flag, we showed that PHLDB3 requires its central domain to bind to MDM2 (Fig. 6e,g). These results demonstrate that PHLDB3 via its central domain can directly interact with the C-terminal domain of MDM2, and also suggest that this protein may act like MDMX to enhance the ability of MDM2 to ubiquitinate p53 without affecting MDM2's auto-ubiquitination and stability, further supporting the functional interaction between PHLDB3 and MDM2 in suppression of p53 functions by mediating p53 proteasomal turnover (Fig. 6h).

**PHLDB3 lessens the chemotherapy sensitivity of HCT116 cells.** Since PHLDB3 was markedly induced after chemotherapeutic drug treatment of p53-containing cancer cells (Fig. 1), we next determined if PHLDB3 has any effect on the chemo-sensitivity of these cancer cells. First, we tested the effect of ectopic PHLDB3 on apoptosis induced by these cytotoxic drugs. Interestingly, ectopic PHLDB3 reduced the basal level of apoptosis of HCT116$^{p53+/+}$ cells and could even significantly downgrade the Doxorubicin- or 5-Fu-induced apoptosis of the cancer cells (Fig. 7a,b) by reducing the protein level of p53 and of its target Puma, which was induced by the two drugs (Fig. 7b), as well as reducing cleaved PARP (Fig. 7b). Next, we carried out a set of cell survival assays by treating either HCT116$^{p53-/-}$ or HCT116$^{p53+/+}$ cells with these two drugs. Compared with the control group, PHLDB3 depletion significantly sensitized HCT116$^{p53+/+}$ cells to 5-Fu and Doxorubicin, with IC$_{50}$ reduced from 13.76 to 6.43 µM for 5-Fu and from 0.07 to 0.03 µM for Dox, respectively (Fig. 7c,d), but did not show a significant effect on the sensitivity of HCT116$^{p53-/-}$ cells to the two drugs (Fig. 7e,f). Again, this result with Doxorubicin was also reproducible in lung cancer cell lines, H460 and H1299

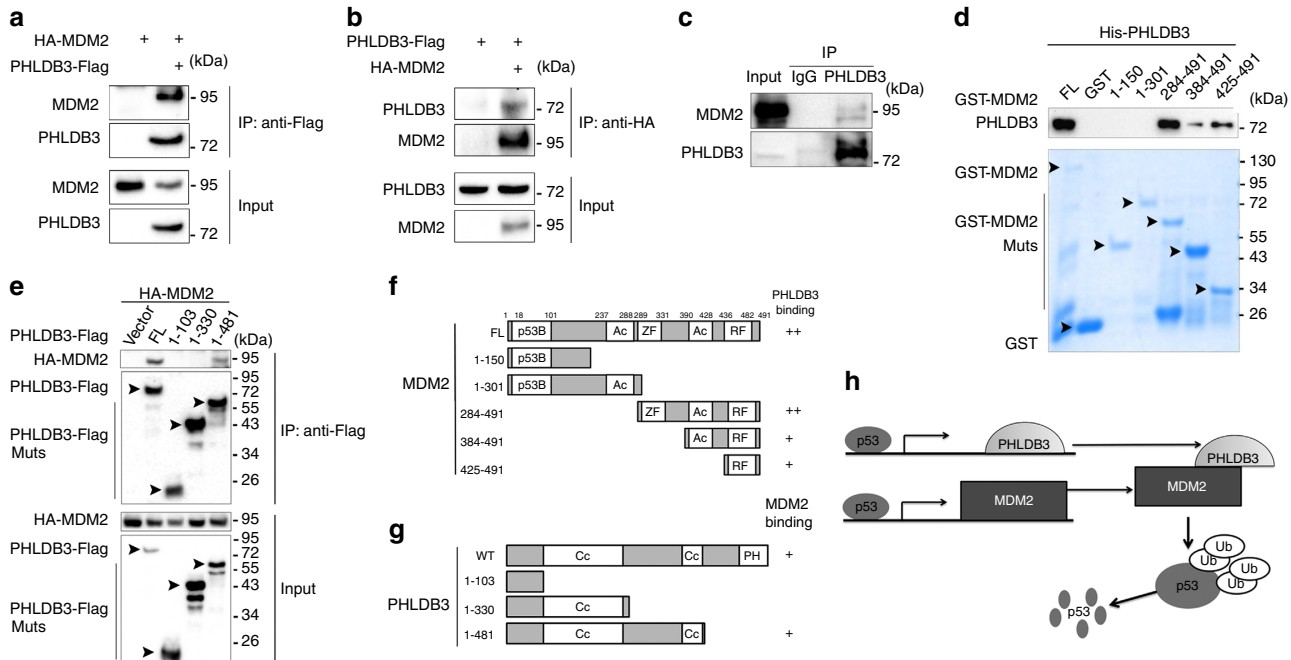

**Figure 6 | PHLDB3 interacts with MDM2.** (**a**,**b**) The interaction between ectopic PHLDB3 and MDM2. HEK293 cells were transfected with plasmids encoding PHLDB3-Flag and/or HA-MDM2 as indicated followed by Co-immunoprecipitation (Co-IP) assays using antibodies as indicated. (**c**) The association between endogenous PHLDB3 and MDM2 is detected after treatment of HCT116$^{p53+/+}$ cells with Doxorubicin. HCT116$^{p53+/+}$ cells were treated with Doxorubicin for 16 h and MG132 for 6 h before harvested for co-IP-IB assays using antibodies as indicated. IgG was used as a control. (**d**) Mapping the PHLDB3 binding domain of MDM2 by GST-pull down assays. Purified GST-tagged MDM2 fragments, including aa 1–150, aa 1–301, aa 284–491, aa 384–491 or aa 425–491, and GST protein alone were incubated with purified His-PHLDB3 for one hour at room temperature. Bound proteins were detected by immunoblotting using anti-PHLDB3 or coomassie staining. (**e**) Mapping the MDM2 binding domain of PHLDB3 by Co-immunoprecipitation. HCT116$^{p53-/-}$ cells were transfected with an HA-MDM2-encoded plasmid along with the plasmid encoding each individual Flag-tagged PHLDB3 fragment as indicated. Co-IP assays were performed using the anti-Flag antibody followed by IB with the anti-HA antibody. (**f**) A schematic diagram of PHLDB3 binding regions on MDM2 based on the result from **d**. (**g**) A schematic diagram of MDM2 binding regions on PHLDB3 based on the result from **e**. (**h**) A model for the negative feedback regulation of p53 by the PHLDB3 and MDM2 complex.

(Supplementary Fig. 2d,e). Taken together, these results suggest that PHLDB3 contributes to the resistance of cancer cells to chemotherapeutic drugs, particularly in wild type p53-containing cancer cells or those cancer cells with amplified PHLDB3, implying that this protein might be a potential molecule target for future development of an anti-cancer therapy.

**PHLDB3 depletion impedes cancer cell growth *in vivo*.** To translate the above-described cellular functions of PHLDB3 into more biological significance, we established a xenograft tumour model by implanting the aforementioned HCT116 (both p53 − / − and p53 + / + ) cell lines that expressed PHLDB3 shRNA or scrambled shRNA into NOD/SCID mice. Tumour growth was monitored for three weeks. As shown in Fig. 8a,b, PHLDB3 depletion more markedly slowed down the growth of xenograft tumour derived from HCT116$^{p53+/+}$ cells than that from HCT116$^{p53-/-}$ cells. Of note, knockdown of PHLDB3 also significantly reduced the growth of tumour derived from HCT116$^{p53-/-}$ cells (Fig. 8a), suggesting that PHLDB3 might also possess a p53-independent activity important for cancer cell growth. At the end of the experiment, the average tumour weight with PHLDB3 depletion was smaller than that with scramble shRNA (Fig. 8c). Consistent with the tumour growth curve, the reduction of tumour weight by PHLDB3 depletion as compared with respective control was more profound in HCT116$^{p53+/+}$ groups (65.9% reduction) than that in HCT116 $^{p53-/-}$ groups (22.8% reduction) (Fig. 8d). To test if PHLDB3 depletion leads to p53 activation, we detected the mRNA and protein levels of p53 and its targets in the xenograft tumours. As anticipated, the

mRNA levels of p21 and PUMA were significantly elevated after PHLDB3 knockdown (Fig. 8e,f). Also, the protein levels of p53 and PUMA were induced in the HCT116$^{p53+/+}$ group, while not changed in the HCT116$^{p53-/-}$ group (Fig. 8g). Collectively, these results demonstrate that PHLDB3 depletion can retard tumour growth by predominantly activating the p53 pathway, though PHLDB3 might also function independently of p53.

## Discussion
Although a number of genes involved in p53 responsive regulation of cell growth and apoptosis have been identified[4,5], here, we report PHLDB3 as an additional p53 transcriptional target gene. Surprisingly, instead of executing p53-dependent suppression of cell growth and proliferation, PHLDB3 counteracts the functions of p53 in a feedback fashion. Because there has not been any single study in literature about the function of PHLDB3, our study as presented here describes the first function of this protein in regulation of p53 and its first biological role in promoting cancer growth. This oncogenic role is also supported by available bioinformatic databases, as the expression profile of PHLDB3 obtained from several bioinformatics database implies that PHLDB3 might play an oncogenic role (Fig. 3)[25–27]. Additionally, by searching the database GenomeRNAi[31], we found out that PHLDB3 knockdown by siRNA is associated with general decreased viability of cancer cells[32] and increased gamma-H2AX phosphorylation[33]. Indeed, our results as shown here demonstrate that PHLDB3 depletion leads to elevated apoptosis (Fig. 4a,b; Supplementary Fig. 2) and increases the sensitivity of

cancer cells to therapeutic drugs (Fig. 7c,d; Supplementary Fig. 2). However, interestingly, these cellular outcomes are more significant when the cancer cells contain wild type p53 (Figs 4 and 7; Supplementary Fig. 2). Consistent with this protumorigenic role, the high mRNA level of PHLDB3 is correlated with worse prognosis of gastric cancer patients, as well as less TP53 mutations (Supplementary Fig. 1; Fig. 3c,d). Furthermore, knockdown of PHLDB3 led to the reduction of xenograft tumours derived from human colorectal cancer HCT116 cells, more significantly in p53-containing cells than in p53-null cells (Fig. 8a–d). Collectively, these results demonstrate that PHLDB3 functions as an oncoprotein and suggest that this oncogenic function might be at least partially p53 dependent.

Our further studies demonstrate that PHLDB3 indeed exerts its protumorigenic effect by restraining p53 activity in a negative feedback manner. First, we showed that the PHLDB3-encoding gene is an authentic p53 target gene (Fig. 2), whose expression at both mRNA and protein levels are p53-dependent in response to

chemotherapeutic agents (Fig. 1). Also, we found that over-expression of PHLDB3 can induce p53 ubiquitination and degradation mediated by MDM2 (Fig. 5), whereas knockdown of PHLDB3 leads to the induction of p53 and its target genes, correspondingly increase of apoptosis and inhibition of cell viability (Fig. 1). However, surprisingly, in human normal lung fibroblast WI-38 cells, Doxorubicin treatment did not induce PHLDB3 expression, despite activation of p53 (Supplementary Fig. 5a). Although this puzzle remains to be solved, we speculate that in normal cells, the epigenetic status of PHLDB3 promoter region might be different from that in cancer cells, or other altered transcription co-regulators present in cancer cells to escort p53 function are missing in normal cells, and therefore p53 is not able to transcriptionally activate PHLDB3. We will further explore this possibility in our future studies. Interestingly, although knockdown of PHLDB3 had a minimal effect on p53 activation (Supplementary Fig. 5b), overexpression of FLAG-PHLDB3 did suppress p53 activity and desensitize p53 response

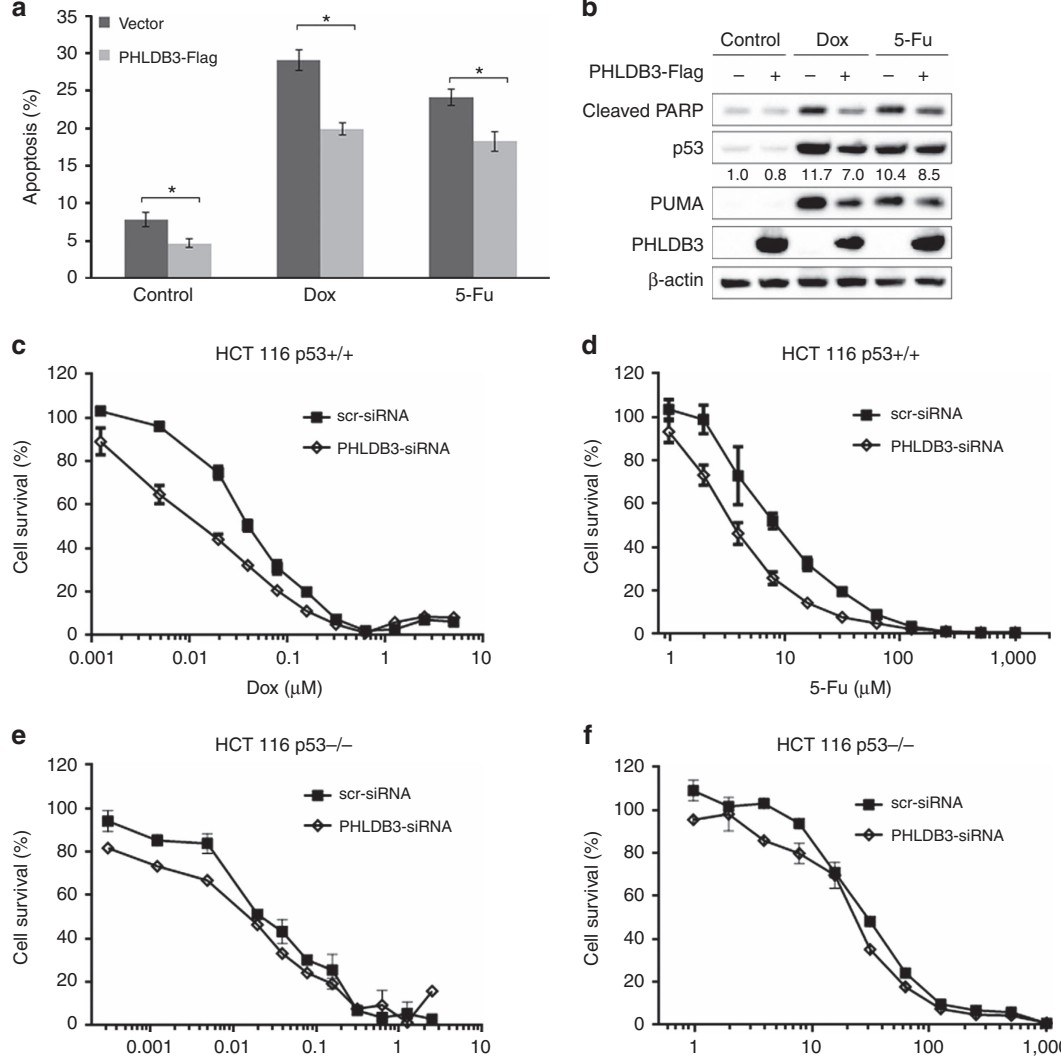

**Figure 7 | PHLDB3 reduces the sensitivity of colorectal cancer cells to chemotherapeutic drugs.** (**a,b**) the effect of PHLDB3 overexpression on apoptosis in HCT116$^{p53+/+}$ following drugs treatment. HCT116$^{p53+/+}$ cells were transfected with PHLDB3-Flag or vector plasmid and then treated with Dox or 5-Fu for 16 h before harvesting. Cells were harvested 48 h post-transfection for flow cytometry analysis (**a**) or immunoblotting with indicated antibodies (**b**). Quantification of Sub-G1 population is shown in **a**. Data represent mean ± s.e.m. of triplicate experiments.*$P < 0.01$ by two-tailed t-test. The relative quantification of p53 protein level is shown in **b**. (**c–f**) Knockdown of PHLDB3 causes more drastic suppression of cell viability in p53-containing cancer cells than in p53-null cancer cells. HCT116$^{p53+/+}$ (**c,d**) or HCT116 p53 $-/-$ (**e,f**) cells were transfected with PHLDB3 or control siRNA, and seeded in 96-well plates next day. Dox or 5-Fu was supplemented for 72 h before cell viability detection by CCK-8.

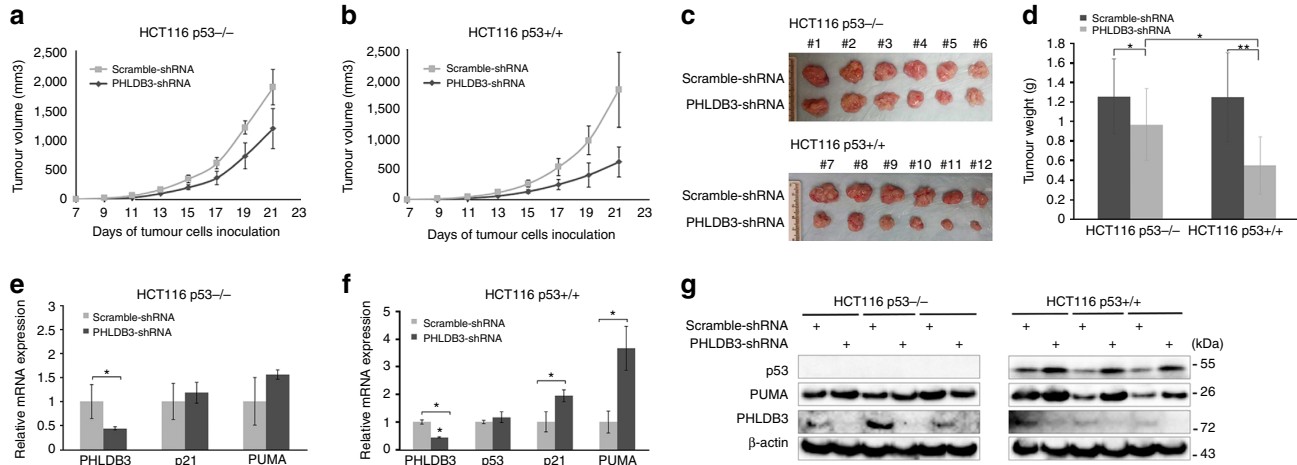

**Figure 8 | PHLDB3 knockdown inhibits tumour growth more significantly by inducing p53.** (**a,b**) Growth curves of xenograft tumours derived from HCT116[p53 +/+] or HCT116[p53 −/−] cells that stably express PHLDB3 shRNA or scramble shRNA. Data are represented as mean ± s.e.m. of mouse xenograft tumours ($n = 6$). (**c**) The images of xenograft tumours collected at the end of the experiment as shown in **a**. (**d**) Graphs for the average weights (mean ± s.e.m., $n = 6$) of harvested xenograft tumours from the above experiments in panels (**a–c**). *$P < 0.05$, **$P < 0.01$ by two-tailed $t$-test. (**e–g**) The mRNA and protein levels of PHLDB3, p53 and p53 targets were assessed in randomly selected three tumour samples by RT-qPCR (**e** and **f**) or immunoblotting analysis with indicated antibodies (**g**). Data in **e** and **f** are represented as mean ± s.e.m. of three randomly selected xenograft tumours; *$P < 0.05$. Data shown in **g** are representative of three individual immunoblotting analyses with antibodies as indicated.

to doxorubicin treatment in WI-38 cells (Supplementary Fig. 5a), indicating that relatively low level of PHLDB3 in normal cells is not sufficient to have dramatic impact on p53, while once overexpressed, consistent with the scenario often found in cancer cells (Fig. 3), PHLDB3 acts as a p53 negative regulator. These intriguing, though preliminary, observations suggest that cancer cells might either remove an inhibitory epigenetic marker present in normal cells or provide a co-activator for p53 absent in normal cells to activate the expression of PHLDB3 as another negative feedback regulator of this tumour suppressor for their growth advantage. This possibility would open a door for an additional direction to p53 and PHLDB3 research to better understand how cancer cells become drug resistance and advance more aggressively even in the presence of wt p53. Mechanistically, PHLDB3 could bind to MDM2 *in vitro* directly and in cells (Fig. 6). This binding appears to be through the central coiled coil domain of PHLDB3 and the C-terminal domain of MDM2 (Fig. 6d–g). Finally, knockdown of PHLDB3 led to the induction of the p53 pathway (Figs 4c–f and 8e–g), consequently the suppression of xenograft tumour growth, more apparently in p53-containing HCT116 cells than in p53-null HCT116 (Fig. 8a–d). Hence, these results convincingly validate that PHLDB3 plays an oncogenic role at least in part by inhibiting the function of p53 via MDM2-mediated degradation in a feedback fashion, and suggest that this protein might be one of the reasons for why some wild type p53-containing cancer cells are more resistant to chemotherapy (Fig. 7)[34,35].

Our studies show that PHLDB3 can enhance MDM2-dependent p53 ubiquitination and proteasomal turnover (Fig. 5) by binding to MDM2. Although the detailed mechanism underlying this regulation remains to be further elucidated, this oncoprotein might do so by enhancing MDM2's ability to ubiquitinate p53 in a way similar to what MDMX does[9], as it binds to the C-terminal domain of MDM2 (Fig. 6e,f), like MDMX[36]. Also, PHLDB3 could not directly affect the stability of p53, as it fails to do so without MDM2 (Fig. 5e). Surprisingly, we observed that ectopic PHLDB3 reduce the formation of the MDM2–MDMX complex, but still binds to the C464A mutant MDM2 that fails to bind to MDMX (Supplementary Fig. 3).

These results suggest that this oncoprotein might replace MDMX, but function similarly to the latter, to enhance MDM2's E3 ubiquitin ligase towards p53, without affecting MDM2 auto-ubiquitination and stability in cancer cells (Fig. 5a). Another potential mechanism is suggested by a high-confidence SIRT1 interactome analysis, showing that PHLDB3 is one of SIRT1-interacting proteins[37]. Since SIRT1 can deacetylate p53 and negatively regulate its protein stability and transcriptional activity[38], it is possible that PHLDB3 might also suppress p53 function by interacting SIRT1 and enhancing its deacetylase activity, consequently facilitating MDM2-mediated p53 ubiquitination and degradation. These possibilities remain to be further investigated.

Intriguingly, we also observed that PHLDB3 could promote cell growth in p53 null HCT116 cells *in vitro* and *in vivo* (Figs 4,7 and 8). This suggests that PHLDB3 might possess a p53-independent oncogenic function as well. Two putative mechanisms may account for this function independent of p53. First, since MDM2 also binds to and suppresses TAp73, a p53 homolog that shares high structural similarity with p53 and transcriptionally activates most of the important p53 target genes, leading to tumour suppression[39,40], it is possible that PHLDB3 may collaborate with MDM2 to inactivate TAp73. The observation that PUMA is moderately induced upon PHLDB3 depletion in p53 null cells (Figs 4g and 8e) also supports this hypothesis. Additionally, PHLDA3, a protein of the pleckstrin homology-like domain family A, can suppress Akt activity as a dominant-negative regulator of Akt and play a prominent role in tumour suppression[20]. Harboring a highly conserved PH domain in the C terminus, PHLDB3 might also crosstalk with PHLDA3 and inhibit its activity, consequently activating the Akt pathway in response to growth signals independently of p53. These speculations are quite tempting and need to be further studied in the near future.

In summary, we present a negative circuit of p53 in cancer cells. Encoded by a p53 target gene, PHLDB3 promotes cancer cell growth *in vitro* and *in vivo*, and induces their resistance to chemotherapeutic drugs by enhancing MDM2-mediated p53 ubiquitination and degradation. Taken together, our study

demonstrates that PHLDB3 functions as an oncoprotein in cancer and could represent a potential therapeutic target in cancers harboring wild type p53, but also in other types of cancers that harbor null or mutant p53.

## Methods

**Plasmids and antibodies.** The PHLDB3 expression plasmid PHLDB3-Flag-Myc was purchased from Origene Technologies, Inc. (cat. no. RC209856). The Flag-tagged plasmids expressing PHLDB3 fragment, aa 1–103, 1–330 or 1–481, were generated by the same approach using the corresponding primers. The two 200-bp p53 binding sites were amplified from human genomic DNA using the primers (RE1, 5′-atatagagctcttcctacccgctgtgtcttc-3′ and 5′-atatactcgagatgaggttcc caacactgg-3′; RE2, 5′-atatagagctctgagacggagtgttgctg-3′ and 5′-atatactcgagaccctg ggtgacagagtgag-3′). The binding sites were subcloned into a promoter-containing pGL3-vector by Sac I and Xho I. The Site-directed mutations of the central p53-binding motif were generated by using the Quickchange Kit (Stratagene). The plasmids PHLDB3 shRNA-1 and -2 were purchased (Sigma-Aldrich, St Louis, MO, USA). The plasmids encoding HA-MDM2, Flag-MDM2, GST-MDM2 fragments, HA-MDMX, p53, Flag-p53, and His-Ub were described previously[41,42]. Anti-Flag (Sigma-Aldrich, catalogue no. F1804, diluted 1:3,000), anti-Cleaved PARP (Cell Signaling Technology, catalogue no. #9541, diluted 1:1,000), anti-Myc (9E10, Santa Cruz Biotechnology, catalogue no. sc-40, diluted 1:1,000), anti-GFP (B-2, Santa Cruz Biotechnology, catalogue no. sc-9996, diluted 1:1,000), anti-PHLDB3 (N-20, Santa Cruz Biotechnology, catalogue no. sc-248246, diluted 1:500), anti-p53 (DO-1, Santa Cruz Biotechnology, catalogue no. sc-126, diluted 1:1,000), anti-p21(CP74, Neomarkers, Fremont, catalogue no. MS-891-P0, diluted 1:1,000), anti-PUMA (H-136, Santa Cruz Biotechnology, catalogue no. sc-28226, diluted 1:1,000) and anti-β-actin (C4, Santa Cruz Biotechnology, catalogue no. sc-47778, diluted 1:3,000) were commercially purchased. Antibodies against MDM2 (2A9 and 4B11) were previously described[41,42].

**Cell culture and transient transfection.** WI-38, H460 and H1299 cells were purchased from American Type Culture Collection (ATCC). HCT116$^{p53+/+}$ and HCT116$^{p53-/-}$ cells were generous gifts from Dr Bert Vogelstein at the John Hopkins Medical Institutes. MEF$^{p53-/-; Mdm2-/-}$ and MEF$^{p53-/-; Mdm2-/-; Mdmx-/-}$ cells were generous gifts from Dr Guillermina Lozano from MD Anderson Cancer Center, the University of Texas, and Dr Jean-Christophe Marine, Vlaams Instituut voor Biotechnologie, Belgium, respectively. STR profiling was performed to ensure cell identity. No mycoplasma contamination was found. All cells were cultured in Dulbecco's modified Eagle's medium (DMEM) supplemented with 10% fetal bovine serum, 50 U ml$^{-1}$ penicillin and 0.1 mg ml$^{-1}$ streptomycin. All cells were maintained at 37 °C in a 5% $CO_2$ humidified atmosphere. Cells seeded on the plate overnight were transfected with plasmids as indicated in figure legends using TurboFect transfection reagent following the manufacturer's protocol (Thermo Scientific). Cells were harvested at 30–48 h post-transfection for future experiments.

**GST fusion protein association assay.** GST-tagged MDM2 fragments were expressed in *E. coli* and conjugated with glutathione-Sepharose 4B beads (Sigma-Aldrich). His-tagged PHLDB3 protein was purified using nickel beads. Purified His-tagged PHLDB3 was incubated and gently rotated with the glutathione-Sepharose 4B beads containing 500 ng of GST-MDM2 fragments or GST only at 4 °C for 4 h. The mixtures were washed three times with GST lysis buffer (50 mM Tris/HCl pH 8.0, 0.5% NP-40, 1 mM EDTA, 150 mM NaCl, 10% glycerol). Bound proteins were analysed by IB with the antibodies as indicated in the figure legends.

**Chromatin immunoprecipitation.** ChIP assay was performed using antibodies as indicated in the figure legends[43]. Briefly, $5 \times 10^7$ H460 cells were seeded in a 15 cm culture dish, treated with DMSO or 1 μM DOX for 24 h and then crosslinked with 1% formaldehyde (37%) at 37 °C for 10 min. Cells were pelleted by centrifugation at 1,000g for 5 min and sonicated to shear DNA to lengths between 200 and 500. Sonicated chromatin was centrifuged at 16,000g for 5 min and then incubated overnight at 4 °C with 1–2 μg antibody against p53 (DO-1, anti-human p53 monoclonal antibody) or normal mouse IgG in the presence of protein A/G-Sepharose beads (Invitrogen). Precipitated chromatin was eluted with 400 μl elution buffer (1% SDS, 0.1 M NaHCO3), incubated at 65 °C for 4 h in the presence of 200 mM NaCl, phenol extracted and precipitated with 3 M sodium acetate and 20 μg of glycogen at −20 °C overnight. RT-qPCR was performed using SYBR Green qPCR Master mix (2 ×) (Thermo Fisher Scientific). The following primer sets were used: RE1 (Fw-5′- TTCCTACCCGCTGTGCTTC-3′, Rv-5′-ATGA GGTTCCCAACACTGG-3′). RE2 (Fw-5′- TGAGACGGAGTGTTGCTG-3′, Rv-5′-GACCCTGGGTGACAGAGTGAG-3′) and p21 (Fw-5′-CTTTCTGGCC GTCAGGAACA-3′, Rv-5′-CTTCTATGCCAGAGCTCAACATGT-3).

**Reverse transcription and quantitative RT-PCR analyses.** Total RNA was isolated from cells using Trizol (Invitrogen, Carlsbad, CA, USA) following the manufacturer's protocol. Total RNAs of 0.5 to 1 μg were used as templates for reverse transcription using poly-(T)$_{20}$ primers and M-MLV reverse transcriptase (Promega, Madison, WI, USA). Quantitative RT-PCR (RT-qPCR) was conducted using SYBR Green Mix according to the manufacturer's protocol (BioRad, Hercules, CA, USA). The primers for PHLDB3, p21, Puma, Mdm2 and GAPDH cDNA detection are as follows: PHLDB3,5′-TGGCCTACTATGCGGACAA-3′ and 5′-GGCGTTCGTAGGTTTTGAG-3′; p53,5′-CCCAAGCAATGGATGATTTGA-3′ and 5′- GGCATTCTGGGAGCTTCATCT-3′; p21, 5′-CTGGACTGTTTTC TCTCGGCTC-3′ and 5′-TGTATATTCAGCATTGTGGGGAGGA-3′; Puma, 5′-ACAGTACGAGCGGCGGAGACAA-3′ and 5′-GGCGGGTGCAGGC ACCTAATT-3′; Mdm2, 5′-ATGAATCCCCCCCTTCCAT-3′ and 5′-CAGGAAG CCAATTCTCACGAA-3′; GAPDH, 5′-GATTCCACCCATGGCAAATTC-3′ and 5′-AGCATCGCCCCACTTGATT-3′.

**Flow cytometry analyses.** Cells transfected with siRNAs as indicated in the figure legends were fixed with ethanol overnight and stained in 500 μl of propidium iodide (Sigma-Aldrich) stain buffer (50 μg ml$^{-1}$ PI, 200 μg ml$^{-1}$ RNase A, 0.1% Triton X-100 in phosphate-buffered saline) at 37 °C for 30 min. The cells were then analysed for DNA content using a BD Biosciences FACScan flow cytometer (BD Biosciences, San Jose, CA, USA). Data were analysed using the CellQuest (BD Biosciences) and Modfit (Verity, Topsham, ME, USA) software programs.

**Cell viability assay.** To assess the long term cell survival, the Cell Counting Kit-8 (CCK-8) (Dojindo Molecular Technologies, Rockville, MD, USA) was used according to the manufacturer's instructions. Cell suspensions were seeded at 2,000 cells per well in 96-well culture plates at 12 h post-transfection. Cell viability was determined by adding WST-8 at a final concentration of 10% to each well, and the absorbance of the samples was measured at 450 nm using a Microplate Reader (Molecular Device, SpecrtraMax M5e, Sunnyvale, CA, USA) every 24 h for 6 days.

**Colony formation assay.** Cancer cells (50% confluence) were transfected with specific or scramble siRNA for 12–18 h and trypsinized. The same number of cells was seeded on each 10-cm plate. Media were changed every 3 days until colonies were visible. Puromycin was added into the media when stable cell lines were used for similar experiments. Cells were then fixed with methanol and stained with crystal violet solution at RT for 30 min. ImageJ was used for quantification of the colonies.

**Immunoblotting.** Cells were harvested and lysed in lysis buffer consisting of 50 mM Tris/HCl (pH7.5), 0.5% Nonidet P-40 (NP-40), 1 mM EDTA, 150 mM NaCl, 1 mM dithiothreitol (DTT), 0.2 mM phenylmethylsulfonyl fluoride (PMSF), 10 μM pepstatin A and 1 mM leupeptin. Equal amounts of clear cell lysate (20–80 μg) were used for immunoblotting (IB) analyses. Each immunoblotting experiment was performed using the same blot, unless otherwise indicated. Uncropped blots are shown in Supplementary Figs 6 and 7.

**Luciferase reporter assay.** H1299 cells were transiently co-transfected with 500 ng p53-responsive luciferase reporter constructs containing either a sequence derived from the identified p53-binding site upstream of the PHLDB3 gene or its mutant sequence (such as RE1 or its mutant as shown in Fig. 2c), 50 ng pRL-TK Renilla reporter (Promega), 500 ng μg pcDNA3.1 and/or FLAG-p53 with Thermo Scientific TurboFect transfection reagent. Cells were lysed for measuring luciferase activity using the Dual-Luciferase Reporter Assay System (Promega) according to the manufacturer's protocol.

***In vivo* ubiquitination assay.** H1299 cells were transfected with plasmids encoding p53, HA-MDM2, His-Ub or PHLDB3-Flag as indicated in the figure legends. At 48 h after transfection, cells were harvested and split into two aliquots, one for IB and the other for ubiquitination assays. Briefly, cell pellets were lysed in buffer I (6 M guanidinium-HCl, 0.1 M Na$_2$HPO$_4$/NaH$_2$PO$_4$, 10 mM Tris-HCl (pH 8.0), 10 mM β-mercaptoethanol) and incubated with Ni-NTA beads (QIAGEN) at room temperature for 4 h. Beads were washed once with buffer I, buffer II (8 M urea, 0.1 M Na$_2$HPO$_4$/NaH$_2$PO$_4$, 10 mM Tris-HCl (pH 8.0), 10 mM β-mercaptoethanol), and buffer III (8 M urea, 0.1 M Na$_2$HPO$_4$/NaH$_2$PO$_4$, 10 mM Tris-HCl (pH 6.3), 10 mM β-mercaptoethanol). Proteins were eluted from beads in buffer IV (200 mM imidazole, 0.15 M Tris-HCl (pH 6.7), 30% glycerol, 0.72 M β-mercaptoethanol, and 5% SDS). Eluted proteins were analysed by IB with the indicated antibodies.

**Immunoprecipitation.** IP was conducted using antibodies as indicated in the figure legends. Briefly, ∼500–1,000 μg of proteins were incubated with the indicated antibody at 4 °C for 4 h or overnight. Protein A or G beads (Santa Cruz Biotechnology) were then added, and the mixture was incubated at 4 °C for additional 1 to 2 h. Beads were washed at least three times with lysis buffer. Bound proteins were detected by IB with antibodies as indicated in the figure legends.

**RNA interference.** siRNAs against PHLDB3 and p53 (Life Technologies, Carlsbad, CA, USA) were commercially purchased. siRNAs (40–60 nM) were introduced into cells using TurboFect transfection reagent following the manufacturer's protocol. Cells were harvested ~72 h after transfection for IB or RT-qPCR.

**Generating stable cell lines.** Lentiviral plasmids based on pLKO.1 were packaged with the 2ndGeneration Packaging System. Briefly, pLKO.1 plasmids containing scrambled or PHLDB3 shRNAs, along with the packaging plasmids pMD2.G and pCMV-dR8.2, were transfected into 293T cells. Cells were maintained at 37 °C in a 5% $CO_2$ humidified atmosphere for 72 h, and the supernatant was harvested and used for infecting HCT116 cells. Media were changed overnight after infection and puromycin ($2\,\mu g\,ml^{-1}$) was added into the media for selection.

**Mouse xenograft experiments.** Seven-week-old female NOD/SCID mice were purchased from Jackson Laboratories. Mice were randomized into two groups (6 mice in each) and subcutaneously inoculated with $5\times10^6$ HCT116 cells that stably expressed scrambled shRNA or PHLDB3 shRNA in the right and left flanks, respectively. Tumour growth was monitored every other day with electronic digital calipers (Thermo Scientific) in two dimensions. Tumour volume was calculated with the formula: tumour volume $(mm^3) =$ (length $\times$ width$^2$)/2. Mice were sacrificed by euthanasia, and tumours were harvested and weighed. To detect p53 activation and apoptotic signals *in vivo*, proteins and RNAs were isolated from tumours either via homogenization in RIPA buffer or Trizol, and then subjected to IB and RT-qPCR analyses. The experiment was not blinded. All of the animals were handled according to approved institutional animal care and use committee (IACUC) protocols (#4275R) of Tulane University School of Medicine. The maximum tumour volume per tumour allowed the IACUC committee is 1.5 cm diameter or $300\,mm^3$ per tumour.

**Statistics.** All *in vitro* experiments were performed in biological triplicate. The Student's two-tailed *t*-test was used to determine mean difference among groups. $P < 0.05$ was considered statistically significant. Data are presented as mean ± s.e.m.

**Data availability.** The authors declare that the data supporting the findings of this study are available within the paper and its Supplementary Information files. The webpages for online genomic and gene expression data bases are mentioned in the legends for Fig. 3, as they are used for this data and the data in Supplementary Fig. 1.

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

## Acknowledgements

We sincerely thank Drs Bert Vogelstein, Guillermina Lozano, and Jean-Christophe Marine for offering cell lines. H.L. was supported in part by NIH-NCI grants R01CA095441, R01CA172468, R01CA127724, R21CA190775, and R21CA201889 as well as the Reynolds and Ryan Families Chair fund. T.C. was supported in part by grant from the National Natural Science Foundation of China (No. 81201820).

## Author contributions

T.C, X.Z., S.X.Z. and H.L. designed the experiments; T.C. conducted most of the studies; X.Z. and H.B.L. conducted part of the Luciferase, ChIP and real-time-PCR analyses; B.C., S.X.Z. and P.L. conducted part of cell proliferation and xenograft analysis; S.X.Z. guided T.C. in xenograft and cell-based studies; Y.C. and H.P prepared and purified recombinant PHLDB3; T.C, X.Z., B.C. and H.L. analysed the data and composed the manuscript.

**Additional information**

**Competing financial interests:** The authors declare no competing financial interests.

**Publisher's note**: 

