## [Peer Review File · Nature Communications]

Reviewer #1 (Remarks to the Author)

In this study, the authors identify a new p53 target gene, PHLDB3, whose protein interacts with Mdm2 and inhibits p53 activity. Mdm2 is required for this inhibition. PHLDB3 knockdown inhibits the growth of xenograft tumors and sensitizes tumor cells to doxorubicin and 5-Fluorouracil. PHLDB3 is overexpressed in some cancers.

Specific comments.

1. Why were the co-IPs performed in HEK293 cells (page 9)? This is the only time this cell line was used.
2. What is the origin of the cell lines used? Have they been authenticated?
3. The data shown in figure 2B does not appear to originate from a single western probed multiple times. The authors need to indicate in the figure legend if the same western was probed in this experiment and the other western experiments shown. If more than one western was used, then actin controls must be shown for all blots.
4. PHLDB3 resembles Mdm2 in that it is a p53 target and in turn targets p53 for proteasomal degradation. Mdm2 is a bonafide p53 inhibitor. Mdm2 should be used for comparison to PHLDB3 in some assays. For example in figure 3, how does Mdm2 knockdown affect apoptosis, cell viability and p53 target gene expression.
5. In figure 5a, the input lanes are as strong as the immunoprecipitation experiment suggesting that all of PHDB3 is binding Mdm2. Is the amount loaded in the input lane a fraction of that used for the experiments?
6. Do tumors with high PHLDB3 show a mutually exclusive relationship with p53 mutation?
7. The manuscript will need some light editing for English.

Reviewer #2 (Remarks to the Author)

In this manuscript, Chao T. et al identified PHLDB3 gene as a novel p53 target gene, which can facilitate MDM2-mediated p53 ubiquitination and degradation. They found that PHLDB3 forms a new negative feedback loop with p53 in cells. Knockdown of PHLDB3 activates p53, which in turn increases apoptosis, inhibits cancer cell proliferation and tumor growth, and sensitizes cancer cells to the chemotherapeutic drugs. Overall, the experiments are well controlled and conducted, and the data are convincing and well-presented. The manuscript is well written. It is a nice study and will further deepen our current understanding of p53 and its regulation in cells.

A few concerns should be addressed to strengthen the findings prior to publication:

1. The results in Fig 1 show that PHLDB3 can be induced by p53 in different cancer cell lines. Data in Fig S1 also suggest the potential role of PHLDB3 in different types of cancer. The results on apoptosis, cell proliferation and drug resistance were mainly obtained from colorectal HCT116 cells. It will be more informative if more cancer cell lines, such as cancer cell lines from different tissues, can be used to test whether these observations made in HCT116 are general in different cancer cell lines.
2. The regulation of PHLDB3 by p53 and the negative regulation of p53 by PHLDB3 were clearly demonstrated in cancer cells, such as HCT116 cells. It will be nice if the authors can test whether these regulations and the negative feedback loop exist in normal human cells or in normal mouse tissues.

Minor points:

1. Fig S1 showed many clinical data which support the potential role of PHLDB3 in different types of tumors (Fig S1). This information is important for readers to understand the physiological significance of PHLDB3 as well as its negative feedback loop of PHLDB3/p53 in cancer. The Fig S1 should be presented as a regular figure in the manuscript instead of a supplemental figure.
2. "quantitative RT-qPCR" should be "RT-qPCR or quantitative RT-PCR".

Reviewer #1:

Specific comments.

1. Why were the co-IPs performed in HEK293 cells (page 9)? This is the only time this cell line was used.

Response: Human embryonic kidney cells (HEK293) have been well established as an efficient host cells to express recombinant proteins¹. Therefore, the interaction between PHLDB3 and MDM2 was initially tested in HEK293 cells by ectopically expressing PHLDB3-FLAG and HA-MDM2. We also confirmed the binding of endogenous PHLDB3 and MDM2 in HCT116^{p53+/+} cells (Fig. 5C).

2. What is the origin of the cell lines used? Have they been authenticated?

Response: We have now included the cell line origins in the Experimental Procedure section. All cell lines have been STR profiling authenticated and cell identity verified by matching STR profiling in ATCC or DSMZ database. If the reviewer need to see the authenticated certificate, we will be happy to show it to him or her.

3. The data shown in figure 2B does not appear to originate from a single western probed multiple times. The authors need to indicate in the figure legend if the same western was probed in this experiment and the other western experiments shown. If more than one western was used, then actin controls must be shown for all blots.

Response: We thank the reviewer for the suggestion. All the bands from Figure 2B were probed from the same blot. We usually cut the blot into several slices so that we could incubate the slices with different antibodies simultaneously. Also, we now included a statement “Each immunoblotting experiment was performed using the same blot, unless otherwise indicated.” in the Experimental Procedure section.

4. PHLDB3 resembles Mdm2 in that it is a p53 target and in turn targets p53 for proteasomal degradation. Mdm2 is a bonafide p53 inhibitor. Mdm2 should be used for comparison to PHLDB3 in some assays. For example in figure 3, how does Mdm2 knockdown affect apoptosis, cell viability and p53 target gene expression.

Response: We thank the reviewer for the comment. The oncogenic properties of MDM2 have been well documented in the literature using both in vitro and mouse model systems. A few studies listed here in the reference²⁻⁵ have demonstrated that MDM2 knockdown inhibits cell proliferation and induces apoptosis. However, the clients of MDM2 are not limited to p53. This could at least partially explain our findings that PHLDB3 is also tumorigenic in p53-deficient cells.

5. In figure 5a, the input lanes are as strong as the immunoprecipitation experiment suggesting that all of PHDB3 is binding Mdm2. Is the amount loaded in the input lane a fraction of that used for the experiments?

Response: We thank the reviewer for this point. Indeed, 5% of the lysates used for IP were loaded as input. However, due to different abundance of input and immunoprecipitated proteins, we adjusted exposure time of the blots, causing the bands appearing comparably strong.

6. Do tumors with high PHLDB3 show a mutually exclusive relationship with p53 mutation?

Response: We thank the reviewer for this important suggestion. We analyzed the relationship between PHLDB3 overexpression and TP53 mutation based on the databases available in cBioPortal, and as shown in Figure 3D and 3E, the event of PHLDB3 gene alteration, mostly upregulation, and that of TP53 mutation, are significantly mutually exclusive, in breast cancer, breast invasive carcinoma and esophageal carcinoma. Data interpretation has been included in the revised manuscript accordingly.

7. The manuscript will need some light editing for English.

Response: We have revised the manuscript carefully.

Reviewer #2:

In this manuscript, Chao T. et al identified PHLDB3 gene as a novel p53 target gene, which can facilitate MDM2-mediated p53 ubiquitination and degradation. They found that PHLDB3 forms a new negative feedback loop with p53 in cells. Knockdown of PHLDB3 activates p53, which in turn increases apoptosis, inhibits cancer cell proliferation and tumor growth, and sensitizes cancer cells to the chemotherapeutic drugs. Overall, the experiments are well controlled and conducted, and the data are convincing and well-presented. The manuscript is well written. It is a nice study and will further deepen our current understanding of p53 and its regulation in cells.

A few concerns should be addressed to strengthen the findings prior to publication:

1. The results in Fig 1 show that PHLDB3 can be induced by p53 in different cancer cell lines. Data in Fig S1 also suggest the potential role of PHLDB3 in different types of cancer. The results on apoptosis, cell proliferation and drug resistance were mainly obtained from colorectal HCT116 cells. It will be more informative if more cancer cell lines, such as cancer cell lines from different tissues, can be used to test whether these observations made in HCT116 are general in different cancer cell lines.

Response: We thank the reviewer for the suggestion. We now included two lung cancer cell lines, H460 and H1299 to test the effect of PHLDB3 in apoptosis, cell proliferation and drug resistance. The new data were presented as Supplementary Figure S2, and the data interpretation was incorporated in the manuscript accordingly.

2. The regulation of PHLDB3 by p53 and the negative regulation of p53 by PHLDB3 were clearly demonstrated in cancer cells, such as HCT116 cells. It will be nice if the authors can test whether these regulations and the negative feedback loop exist in normal human cells or in normal mouse tissues.

Response: We agree with the reviewer that it is important to test the feedback regulation between PHLDB3 and p53 in normal cells. We have now included additional experiments using human lung fibroblast WI-38 cells as Supplementary Figure S5 and the data discussion was incorporated into the manuscript accordingly. Surprisingly, but interestingly, this PHLDB3-p53 feedback loop was specific to cancer cells, which suggests that cancer might either utilize an epigenetic mechanism to turn on the p53RE-containing promoter of PHLDB3 or recruit an oncogenic co-activator of p53 to activate the expression of PHLDB3 at the transcriptional level in order to inactivate p53 via this newly identified p53 suppressor in cancer cells that harbor wild type p53, although these two conjectures need to be further investigated. This opens a new direction for this research.

Minor points:

1. Fig S1 showed many clinical data which support the potential role of PHLDB3 in different types of tumors (Fig S1). This information is important for readers to understand the physiological significance of PHLDB3 as well as its negative feedback loop of PHLDB3/p53 in cancer. The Fig S1 should be presented as a regular figure in the manuscript instead of a supplemental figure.

Response: We now have moved previous Fig S1 into main figures as Figure 3A, 3B and 3C, and changed the order of other figures accordingly.

2. "quantitative RT-qPCR" should be "RT-qPCR or quantitative RT-PCR".

Response: We thank the reviewer for pointing out the typo and now corrected it accordingly.

Reference

- 1 Thomas, P. & Smart, T. G. HEK293 cell line: a vehicle for the expression of recombinant proteins. *J Pharmacol Toxicol Methods* **51**, 187-200, doi:10.1016/j.vascn.2004.08.014 (2005).
- 2 de Rozieres, S., Maya, R., Oren, M. & Lozano, G. The loss of mdm2 induces p53-mediated apoptosis. *Oncogene* **19**, 1691-1697, doi:10.1038/sj.onc.1203468 (2000).
- 3 Langheinrich, U., Hennen, E., Stott, G. & Vacun, G. Zebrafish as a model organism for the identification

- and characterization of drugs and genes affecting p53 signaling. *Curr Biol* **12**, 2023-2028 (2002).
- 4 Mu, Z. *et al.* Antisense MDM2 enhances the response of androgen insensitive human prostate cancer cells to androgen deprivation in vitro and in vivo. *Prostate* **68**, 599-609, doi:10.1002/pros.20731 (2008).
- 5 Zhang, Z. *et al.* Stabilization of E2F1 protein by MDM2 through the E2F1 ubiquitination pathway. *Oncogene* **24**, 7238-7247, doi:10.1038/sj.onc.1208814 (2005).

Reviewer #1 (Remarks to the Author)

- A. the authors have identified a new p53 inhibitor.
- B. novel
- C. increased quality of data in revision
- D. yes
- E. the new data in figure 3 are very convincing
- F. none - great responses to criticism
- G. yes
- H. yes

Reviewer #2 (Remarks to the Author)

The authors have addressed my comments. The manuscript has been improved substantially. I recommend the acceptance of this manuscript.